# Loss of Concentration May Occur by Blink Inhibition in DED Simulation Models

**DOI:** 10.3390/life10050061

**Published:** 2020-05-13

**Authors:** Yasue Mitsukura, Kazuno Negishi, Masahiko Ayaki, Mayuko Santo, Motoko Kawashima, Kazuo Tsubota

**Affiliations:** 1Department of Technology and Engineering, Keio University, Yokohama 2238532, Japan; mitsukura@sd.keio.ac.jp; 2Department of Ophthalmology, Keio University School of Medicine, Tokyo 1608582, Japan; motoko326@gmail.com (M.K.); tsubota@z3.keio.jp (K.T.); 3Otake Clinic Moon View Eye Center, Yamato 242001, Japan; 4Keio University School of Medicine, Tokyo 1608582, Japan; mayuko0310@keio.jp

**Keywords:** dry eye disease, electroencephalogram, inter-blinking time, evaporative dry eye

## Abstract

Purpose: Patients with dry eye disease (DED) often suffer productivity loss and distress due to bothersome symptoms. The aim of this study was to objectively quantify and compare productivity-related emotional states obtained from brain waveforms in natural and simulated DED conditions. Method: 25 healthy adults (6 females and 19 males; mean age ± standard deviation, 22.6 ± 8.3 years) were recruited for the study, which included an electroencephalogram (EEG), measurements of interblinking time, and questionnaires. DED was simulated by suppressing blinking, while spontaneous blinking served as a control. Elements of concentration, stress, and alertness were extracted from the raw EEG waveforms and the values were compared during spontaneous and suppressed blinking. The relation with DED-related parameters was then explored. Written informed consent was obtained from all participants. Results: All participants successfully completed the experimental protocol. Concentration significantly decreased during suppressed blinking in 20 participants (80%), when compared with spontaneous blinking, whereas there were no or small differences in stress or alertness between spontaneous and suppressed blinking. The change in concentration was correlated with interblinking time (β = −0.515, *p* = 0.011). Conclusion: Loss of concentration was successfully captured in an objective manner using the EEG. The present study may enable us to understand how concentration is affected during blink suppression, which may happen in office work, particularly during computer tasks. Further study using detailed ocular evaluation is warranted to explore the effect of different interventions.

## 1. Introduction

Dry eye disease (DED) is a multifactorial ocular surface disorder with symptoms including abnormal tear functions, keratoepitheliopathy, and ocular discomfort [1]. The overall prevalence of DED is around 20–40%, and it is most prevalent in older women and the Asian population [2,3]. Common nonvisual subjective symptoms of DED include dryness, irritation, pain, and heaviness of the eyelids. DED also has visual symptoms, including eye fatigue, blurring, and photophobia, which may be annoying during working, reading, and various visual activities [4,5]. Quality of life [6,7] and subjective happiness [8] may also deteriorate with DED. In addition to the reported nonocular comorbidities, including depression and sleep disorders [9,10], productivity loss has been suggested in office workers with DED [11,12,13,14,15,16]. Symptomatic DED with bothersome symptoms is often suggested to be associated with productivity loss, rather than diagnostic DED. Potential symptoms associated with productivity loss may be partly derived from the trigeminal nerve, with patients even suffering difficulty in keeping their eyes open due to pain and dryness [17,18,19,20,21]. However, to date, there have been few studies evaluating the neuropsychiatric status of patients with DED by objective examination of DED-related symptoms, except for functional magnetic resonance imaging (fMRI) [22] and functional near-infrared ray spectroscopy [23]. Tsubota et al. [22] found normal blinking activated areas in the orbitofrontal cortex and in some cases, the visual cortex. In severe DED, blink inhibition strongly activated the visual cortex even after irritation was removed by topical anesthesia. Ono et al. [23] found that low prefrontal cortex activation was detected in normal controls without ocular discomfort, and high activation was detected in both control and DED participants with ocular discomfort. Prefrontal cortex activity was confirmed with ocular discomfort when the eyes were open, decreased with lubricant, and almost disappeared with anesthetic for all participants.

The KANSEI analyzer (Mindwave; NeuroSky Inc., San Jose, CA and Dentsu ScienceJAM Inc., Tokyo, Japan) has been developed for measuring particular elements of human emotions or states of mind, such as concentration, stress, alertness, like, and interest. It is a wireless, one-channel system with an Fp1 channel and a reference ear clip, A1, allowing for the recording of a resting electroencephalogram (EEG) with minimal noise, despite being prone to artifacts due to eye blinks or muscle movement in the frontal region with eye opening. This instrument was recently recognized and validated for clinical use [24]. For example, the algorithm used for the concentration element was developed to analyze the EEGs recorded from over 100 participants during calculation contests and computer animation battle games and the results were comparable with those for calculation task. Alertness [25] and stress [26] can be also evaluated with an algorithm developed for each emotion. The KANSEI analyzer is commercially available and widely used for measuring consumer preferences for products and advertisements, driving safety, and more. The advantage of the wireless EEG is that it is acquired in a real-time setting, and is superior to questionnaires and other conventional methods, including saliva amylase and fMRI. Yet, the EEG contains electromyogram (EMG) event-related potentials and interferences, and the present experiment involved blinking that evoked noise in the EEG. However, the originally developed algorithm eliminated and minimized EMG and other noise as much as possible. Preparatory calibration in advance of the experiment determined the baseline EEG for each participant in order to extract the specific wave.

Evaporative DE is a major subtype of DED, along with short tear break-up time DED. Evaporative DED is characterized as an excessive evaporation of tears and can be clinically simulated with wind, a dry environment, or blink suppression [27,28]. In the present study, blink suppression was used to simulate evaporative DED, based on an animal model established by Nakamura and colleagues [27,28]. Their system is made of a suspended cylinder hanging on a swing, in constant air flow, where the rats were placed and forced to keep their eyes open to prevent falling. The aim of the present study was to objectively quantify the change in EEG in normal and simulated DED, and to compare these two conditions. The present study may enable us to capture how DED patients can be disabled during office work, as well as any other activities that use vision.

## 2. Methods

### 2.1. Study Design, Ethical Approval, and Study Population

The institutional review board and ethics committee of the Department of Technology and Engineering, Keio University (permit number 31-9; approved, 2 April 2019) approved this study, and the study was conducted in accordance with the tenets of the 1995 Declaration of Helsinki (as revised in Edinburgh, 2000). Informed written consent for study participation was obtained from study participants after sufficient explanation by a principal investigator or adequately educated surrogate following approved protocol. This study was registered through the University Hospital Medical Information Network Center on 1 September 2019 (number UMIN000037827).

### 2.2. Study Participants and Screening Procedures

Healthy adults were recruited via an announcement in a medical facility. Potential participants were screened for an annual health check-up conducted at their office or school. Participants with chronic medical or psychological conditions, sleep disorders, and those taking prescription medications were excluded from the study. Participants with a history of any ocular surgery or shift work within the previous week were also excluded. Participants were instructed to refrain from using contact lenses or ophthalmic medications on the day of the experiment.

### 2.3. Electroencephalogram–KANSEI Analyzer

Developed by one of the authors (YM), the KANSEI analyzer (Figure 1) detects and extracts the elements of concentration, stress, alertness, like, and interest from the EEG recording [24]. Each element is extracted according to a calculation derived from the accumulated standard data recorded from over 100 examinees. 

To record an EEG with the KANSEI analyzer, the participants wear a headset and put an electrode on their forehead and earlobe. The protocol consisted of a 30- to 90-s period of calibration, during which the participants stayed calm with their eyes closed. Then, they were instructed to look straight ahead with normal blinking for 60 s, then to keep their eyes open for 60 s but were allowed to blink when they felt uncomfortable. The final task was spontaneous blinking after the suppressed blink. Soft blinking did not affect the meibomian gland, tear film, or EMG. All experiments were performed indoors at a temperature of 18–25 °C, humidity of 40–60%, with illumination below 1000 lux.

### 2.4. Questionnaires and Interblinking Time

The participants were asked to measure their interblinking time (IBT) and complete a Dry Eye-Related Quality-of-Life Score (DEQS) questionnaire [29,30] and a short questionnaire [31]. IBT was measured as the time (in seconds) eyes were kept open for as long as possible, and the suggested cutoff value for symptomatic DED was 10 [32]. DEQS is a 14-item self-administered questionnaire to measure the severity of bothersome symptoms in DED and the suggested cutoff value was 15 [28]. The short questionnaire comprised of two symptom questions: (1) “How often do your eyes feel dry?” and (2) “How often do your eyes feel irritated?” The third question was as follows: “Have you ever been diagnosed as having DE syndrome?”

### 2.5. Statistical Analysis

We analyzed the recorded amplitude of the extracted elements of the brainwave for concentration, stress, and alertness during a 30-s period when the waveform seemed stable, after the initiation of the assigned tasks of spontaneous blinking and suppressed blinking for 60 s. We used paired *t*-tests for comparison between spontaneous blinking and suppressed blinking in an individual. To assess the association of the results of the questionnaire with the EEG values, Spearman correlation coefficients were calculated. The raw value of the recorded amplitude varied according to baseline; thus, we calculated the difference in EEG amplitude during spontaneous blinking and suppressed blinking. All statistical tests were two-sided, and the significance level was set to an α of 0.05. All analyses were performed using STATFLEX software (Atech, Osaka, Japan).

## 3. Results

Twenty-five healthy adults (6 females and 19 males; mean age ± standard deviation, 22.6 ± 8.3 years) successfully completed the protocol of EEG recording and questionnaires. The mean DEQS score was 14.7 ± 17.5 and the mean IBT was 49.7 ± 51.0 s. The proportion of participants with symptomatic DED was 28.0% (n = 7) according to the DEQS, 12.0% (n = 3) according to the short questionnaire, and 4.0% (n = 1) according to IBT. 

A representative brainwave showing the extracted elements of concentration, stress, and alertness is shown in Figure 2. Concentration significantly decreased during suppressed blinking in 20 participants (80%), compared with spontaneous blinking, and this decrease was more than 10% in 48.0% of the participants (n = 12) during blink suppression (*p* < 0.001, paired *t-*test; Table 1). This value recovered by more than 10% in 60.0% of the participants (n = 15) during the second session of spontaneous blinking (*p* < 0.001, paired *t*-test). Stress significantly decreased more than 10% in 36.0% of the participants (n = 9) during blink suppression (*p* < 0.001, paired *t*-test) and recovered more than 10% in 24.0% of participants (n = 6) during the second session of spontaneous blinking (*p* = 0.032, paired *t*-test). The increase in concentration after the second session of spontaneous blinking was 12.8% ± 10.2% and was much greater than the increase in stress (4.6% ± 10.0%; *p* < 0.001). Alertness did not change significantly from spontaneous to suppressed blinking (1.8% ± 9.0%), or from suppressed to spontaneous blinking (2.4% ± 2.6%).

Brainwave elements for concentration (red), stress (green), and alertness (black) extracted from the EEG recording during spontaneous blinking (Spont) and suppressed blinking (Supp) are shown. The EEG was continuously recorded during the tasks of Spont, Supp, and Spont, from left to right. Each task was assigned for 60 s. This participant showed decreased concentration values in the Supp phase, compared to the Spont phase. The changes in stress and alertness were smaller than those for concentration.

The delta concentration was correlated with IBT (β = −0.515; *p* = 0.011; Table 2 and Figure 3), whereas there was no correlation between delta stress and IBT. The delta alertness tended to be associated with IBT (β = 0.385; *p* = 0.054; Figure 4).

The decrease in concentration between spontaneous and suppressed blinking was negatively correlated with IBT (β = −0.515; *p* = 0.011).

The increase in alertness between spontaneous and suppressed blinking was weakly associated with IBT (β = 0.385; *p* = 0.054).

## 4. Discussion 

The KANSEI analyzer successfully captured the difference in the state of mind between spontaneous blinking and suppressed blinking. According to previous investigations, there may be local and central events involved in blinking and three possible roles of blinking could be involved in our observations: (1) keeping a healthy ocular surface for optical performance [32,33,34,35]; (2) relieving eye fatigue by filtering blue light irradiation into the eye [36,37,38,39,40,41]; and (3) vision suppression [42,43,44]. In the first role, blinking keeps the ocular surface wet and smooth by interrupting evaporation and wiping the ocular surface to spread tear components. Blink suppression is a hard task for DED patients since they tend to blink more frequently than people without DED [32,33]. Acute ocular surface drying activates the stress-signaling circuit on the ocular surface and in immune cells, as clinical and experimental studies have revealed that desiccation is a potent stress to the ocular surface that initiates a secondary immune response [35]. Blinking can also relieve and prevent a variety of ocular surface symptoms, such as dryness, irritation, pain, lacrimation, and blurring. It is a reasonable expectation that concentration may decrease during blink suppression while the participants tolerate increasing discomfort.

Secondly, blinking and eye closures function as photoprotection for preventing ocular surface [36,37] and retinal [38] damage, and relieving eye fatigue [39,40,41] by reducing ultraviolet and blue light irradiation into the eye. We previously proposed a hypothesis [40] that trigeminal activation might occur in conditions where photophobia/photoallodynia is a presenting symptom of eye fatigue, involving systems that alter melanopsin-based signaling from intrinsically photosensitive retinal ganglion cells [41]. Suppression of blinking disturbs the photoprotective defense reaction and may weaken concentration.

The third role of blinking is associated with brain activation for visual attention. fMRI studies have demonstrated that visual perception is actively reduced during blinking [42,43] and suppressed blinking may activate busy neural networks. Berman et al. [44] investigated blink suppression by analyzing fMRI data to detect brain activation associated with the buildup of the urge to blink. They found a widespread network of brain activation, including in the insular cortex, which helps support the buildup of urge in blink suppression. Therefore, it is likely that concentration might decrease under busy brain activation. Two previous studies indicated an incomplete response to topical anaesthesia for suppressed blinking [22] and pain [45] in DED patients, and these observations support our second and third hypotheses proposing that blinking in evaporative DED patients may be induced by factors other than the ocular surface. Further investigations are warranted to confirm the present results since blinking is associated with numerous physical, neuronal, and mental factors [43].

The positive correlation between IBT and the decrease in concentration strongly supports our first two hypotheses for local events. The brain may be activated in every participant with or without DED and the third role may be weaker than local events. Taken together, the present results successfully show that productivity loss may occur in evaporative DED due to decreased concentration. Considering the high prevalence of the short tear break-up time DED type in Japan [2,46] and its clinical impact on office workers with productivity loss [11], DED care and treatment could be recommended to improve workers’ health and increase productivity. Most participants exhibited a decrease in concentration during blink suppression and this result could be applicable to the general population, since between 4.0% and 28.0% of participants in the present study had symptoms of DED, as determined by questionnaires or IBT.

The stress element of the brainwave significantly decreased during blink suppression; however, we speculate this change may not be the same phenomenon as concentration, since the magnitude was much smaller than the change in concentration and we did not find a consistent response during spontaneous, suppressed, and spontaneous blinking. Alertness did not exhibit any remarkable change, and this should be further confirmed with a longer experimental session, although increases in alertness from spontaneous to suppressed blinking were reasonably associated with IBT suggesting alertness of DED patients may decrease during computer work leading to productivity loss.

The present study has several limitations. First, this study lacks results from patients with formally diagnosed DED. Further study with larger volumes of participants with and without DED would confirm the present results by establishing the real application of this methodology by comparing the results obtained from DED and simulated conditions. Detailed ocular evaluations for DED patients with mild, moderate and severe conditions should be contributed to evaluate the applicability and suitability of the present results, in addition to validated questionnaires and IBT to estimate the DED condition of participants we used. Second, further investigation of EEG is warranted under different conditions, such as the use of topical anesthesia and lubricant, wind, humidity, and other possible interventions relieving or worsening the condition of the ocular surface. Finally, the present experimental model is not the same as in DED patients since blink suppression itself affects EEG and EMG. More suitable experimental models should be established for spontaneous blinking with excessive evaporation or tear instability.

## Figures and Tables

**Figure 1 life-10-00061-f001:**
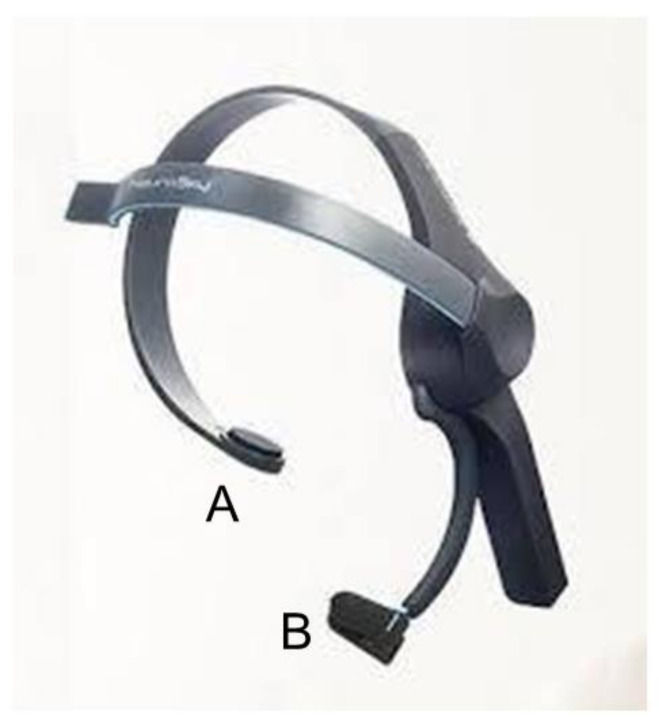
KANSEI analyzer. The KANSEI analyzer is a one-channel electroencephalogram designed to extract human emotions, including stress, concentration, alertness, like, and interest. Electrodes are set on the forehead (**A**) and left earlobe (**B**).

**Figure 2 life-10-00061-f002:**
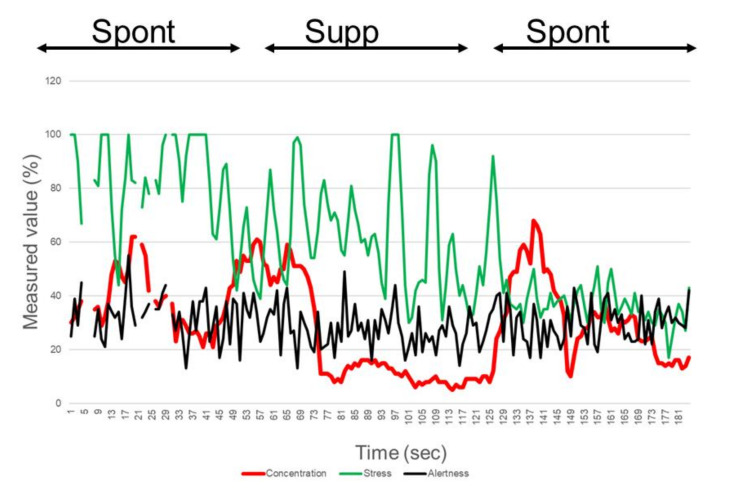
Electroencephalogram (EEG) recording of a representative participant (20 y/o, female) using the KANSEI analyzer.

**Figure 3 life-10-00061-f003:**
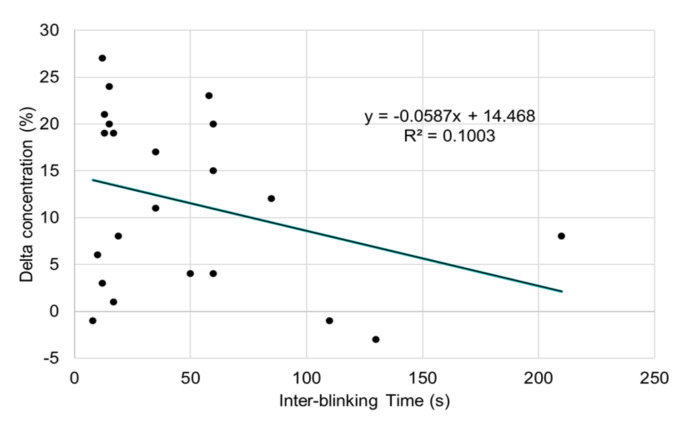
Scatter plots of delta concentration and interblinking time (IBT).

**Figure 4 life-10-00061-f004:**
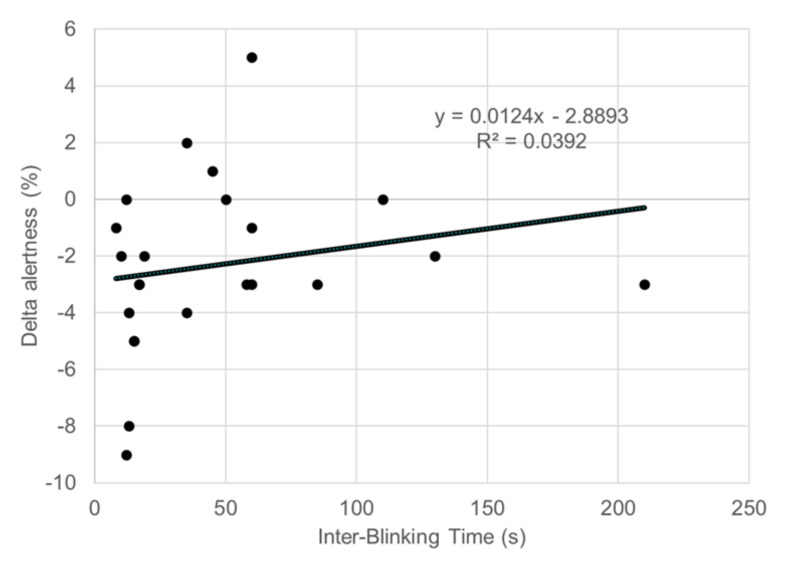
Scatter plots of delta alertness and interblinking time (IBT).

**Table 1 life-10-00061-t001:** Concentration, stress, and alertness elements during spontaneous blinking (Spont) and suppressed blinking (Supp).

	Concentration			Stress			Alertness		
	1 Spont	2 Supp	3 Spont	1 Spont	2 Supp	3 Spont	1 Spont	2 Supp	3 Spont
Mean (%)	43.88 ± 10.20	34.32 ± 12.78	47.12 ± 9.09	34.64 ± 13.97	24.64 ± 9.53	29.20 ± 9.58	70.92 ± 6.05	72.72 ± 6.24	70.36 ± 6.04
Paired *t*-test *p* values	<0.001 (vs. 2)	<0.001 (vs. 3)	0.045 (vs. 1)	<0.001 (vs. 2)	0.032 (vs. 3)	0.069 (vs. 1)	0.391 (vs. 2)	0.732 (vs. 3)	0.964 (vs. 1)

Data are shown as mean ± standard deviation.

**Table 2 life-10-00061-t002:** Regression analysis of electroencephalographic results and dry eye-related parameters.

	Delta Concentration		Delta Stress		Delta Alertness	
	β	*p* value	β	*p* value	β	*p* value
Non-adjusted						
IBT	−0.381	0.066	−0.130	0.552	0.378	0.062
DEQS	0.002	0.991	−0.150	0.473	−0.033	0.874
Symptomatic dry eye	−0.297	0.157	−0.086	0.687	0.033	0.874
Adjusted for age and sex						
IBT	−0.515	0.011*	−0.139	0.524	0.385	0.054
DEQS	0.022	0.923	−0.222	0.331	−0.086	0.696
Symptomatic dry eye	−0.255	0.243	−0.065	0.772	0.118	0.584

**p* < 0.05. Delta Concentration, delta Stress, and delta Alertness are the electroencephalographic values (%) for (spontaneous blink- suppressed blink). Abbreviations: IBT, interblinking time; DEQS, Dry Eye-Related Quality-of-Life Scale.

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
