# Peer review of "Loss of Concentration May Occur by Blink Inhibition in DED Simulation Models"

_life, 2020, doi:10.3390/life10050061_

Round 1
Reviewer 1 Report
In this study the authors try to quantify and compare productivity-related emotional states obtained from brain waveforms in natural and simulated
dry eye disease conditions.The KANSEI analyzer successfully captured the difference in state of mind between spontaneous blinking and suppressed blinking. The major lag in this study is the lack of results from patients with DED conditions rather we have it in simulated conditions. It will be of major importance if the authors can show some results from DED patients so that we can understand the the real application of this study. The study fails to establish the importance of the application.
(i) The authors should include data fron DED patients
(ii) If possible from mild, moderate and severe conditions.
Extra comments: As I mentioned in my comment, the major flaw is the simulation of dry eye conditions in normal patients to study the technique. It fails to support the application in dry eye disease condition.
The aim of this study should objectively quantify and compare 16 productivity-related emotional states obtained from brain waveforms in natural and DED conditions rather than simulated condition. Especially should have been tested in patients with mild to severe DED conditions and this can be compared with simulated DED. The authors have concluded that further study using detailed ocular evaluations is warranted to explore the effect of different interventions and this should be done in real disease condition rather than simulated condition.Author Response
Response to Reviewer 1 Comments
Thank you very much for reviewing our manuscript. To aid in the re-review of this manuscript, we have included a point-by-point response to each comment. The reviewer’s comments are italicized and placed in square brackets. In addition, within the revised manuscript, we have used underlined text to highlight changes in response to the reviewers’ comments.
We appreciate the suggestions and comments by the reviewer. As a consequence of valuable suggestions, we believe that our manuscript has been much improved.
[Point 1: In this study the authors try to quantify and compare productivity-related emotional states obtained from brain waveforms in natural and simulated dry eye disease conditions. The KANSEI analyzer successfully captured the difference in state of mind between spontaneous blinking and suppressed blinking. The major lag in this study is the lack of results from patients with DED conditions rather we have it in simulated conditions. It will be of major importance if the authors can show some results from DED patients so that we can understand the the real application of this study. The study fails to establish the importance of the application.
(i) The authors should include data fron DED patients
(ii) If possible from mild, moderate and severe conditions.]
We appreciate the reviewer’s complimentary comments. To improve our manuscript, we have amended the limitation section as follows.
[Discussion, page 8]
“The present study has several limitations. First, this study is the lack of results from patients with formally diagnosed DED. Further study with large volume of participants with and without DED would confirm the present results to establish the real application of this methodology by comparing the results obtained from DED and simulated conditions. Detailed ocular evaluations for DED patients with mild, moderate and severe conditions should be contributory to evaluate the applicability and suitability of the present results in addition to validated questionnaires and IBT to estimate DED condition of participants we used. Second, further investigation of EEG is warranted under different conditions, such as the use of topical anesthesia and lubricant, wind, humidity, and other possible interventions relieving or worsening the condition of the ocular surface. Finally, the present experimental model is not the same as in DED patients since blink suppression itself affects EEG and EMG. More suitable experimental models should be established for spontaneous blinking with excessive evaporation or tear instability. “
[Point 2: Extra comments: As I mentioned in my comment, the major flaw is the simulation of dry eye conditions in normal patients to study the technique. It fails to support the application in dry eye disease condition. The aim of this study should objectively quantify and compare 16 productivity-related emotional states obtained from brain waveforms in natural and DED conditions rather than simulated condition. Especially should have been tested in patients with mild to severe DED conditions and this can be compared with simulated DED. The authors have concluded that further study using detailed ocular evaluations is warranted to explore the effect of different interventions and this should be done in real disease condition rather than simulated condition.]
We appreciate the reviewer’s complimentary comments. We already responded this comment before.
Reviewer 2 Report
This study may provide ophthalmologists and dry eye patients to understand how concentration is affected during blink suppression, which may happen in office work, particularly during computer. Loss of concentration was successfully captured in an objective manner using the EEG. Further study using detailed ocular evaluations is warranted to explore the effect of different interventions.
Author Response
Response to Reviewer 2 Comment
Thank you very much for reviewing our manuscript. To aid in the re-review of this manuscript, we have included a point-by-point response to each comment. The reviewer’s comments are italicized and placed in square brackets. In addition, within the revised manuscript, we have used underlined text to highlight changes in response to the reviewers’ comments.
We appreciate the suggestions and comments by the reviewer. As a consequence of valuable suggestions, we believe that our manuscript has been much improved.
[This study may provide ophthalmologists and dry eye patients to understand how concentration is affected during blink suppression, which may happen in office work, particularly during computer. Loss of concentration was successfully captured in an objective manner using the EEG. Further study using detailed ocular evaluations is warranted to explore the effect of different interventions.]
We appreciate the reviewer’s complimentary comments. To improve our manuscript, we have amended the limitation section as follows.
“The present study has several limitations. First, this study is the lack of results from patients with formally diagnosed DED. Further study with large volume of participants with and without DED would confirm the present results to establish the real application of this methodology by comparing the results obtained from DED and simulated conditions. Detailed ocular evaluations for DED patients with mild, moderate and severe conditions should be contributory to evaluate the applicability and suitability of the present results in addition to validated questionnaires and IBT to estimate DED condition of participants we used. Second, further investigation of EEG is warranted under different conditions, such as the use of topical anesthesia and lubricant, wind, humidity, and other possible interventions relieving or worsening the condition of the ocular surface. Finally, the present experimental model is not the same as in DED patients since blink suppression itself affects EEG and EMG. More suitable experimental models should be established for spontaneous blinking with excessive evaporation or tear instability.“
Round 2
Reviewer 1 Report
It is good that the authors have added a paragraph about the pitfalls in the study. Since the authors were not able to add patient data, they should change the title as "Loss of concentration may occur by blink inhibition in DED simulation models" so that it send the correct message to the readers.
Author Response
Response to Reviewer Comments
Thank you very much for reviewing our manuscript. We have amended the manuscript accordingly.
[Comment: It is good that the authors have added a paragraph about the pitfalls in the study. Since the authors were not able to add patient data, they should change the title as "Loss of concentration may occur by blink inhibition in DED simulation models" so that it send the correct message to the readers.]
We have amended the title accordingly.
This manuscript is a resubmission of an earlier submission. The following is a list of the peer review reports and author responses from that submission.